# Could changing invitation and booking processes help women translate their cervical screening intentions into action? A population-based survey of women's preferences in Great Britain

Mairead Ryan, Jo Waller, Laura AV Marlow

Department of Behavioural Science and Health, University College London, London, UK

**Correspondence to**
Laura AV Marlow;
l.marlow@ucl.ac.uk

## ABSTRACT

**Objectives** Many women who do not attend screening intend to go, but do not get around to booking an appointment. Qualitative work suggests that these 'intenders' face more practical barriers to screening than women who are up-to-date ('maintainers'). This study explored practical barriers to booking a screening appointment and preferences for alternative invitation and booking methods that might overcome these barriers.
**Design** A cross-sectional survey was employed.
**Setting** Great Britain.
**Participants** Women aged 25–64, living in Great Britain who intended to be screened but were overdue ('intenders', n=255) and women who were up-to-date with screening ('maintainers', n=359).
**Results** 'Intenders' reported slightly more barriers than 'maintainers' overall (mean=1.36 vs 1.06, t=3.03, p<0.01) and were more likely to think they might forget to book an appointment (OR=2.87, 95% CI: 2.01 to 4.09). Over half of women said they would book on a website using a smartphone (62%), a computer (58%) or via an app (52%). Older women and women from lower social grades were less likely to say they would use online booking methods (all ps <0.05). Women who reported two or more barriers were more likely to say they would use online booking than women who reported none (ps <0.01).
**Conclusions** Women who are overdue for screening face practical barriers to booking appointments. Future interventions may assess the efficacy of changing the architecture of the invitation and booking system. This may help women overcome logistical barriers to participation and increase coverage for cervical screening.

### Strengths and limitations of this study

► This was the first study to break down the invitation and booking process into its component parts, identifying barriers at each stage of the process and alternative booking options which may help women to overcome these barriers.
► Women were purposely recruited to be up-to-date and overdue; however, response rate was not recorded.
► The practical barriers cited in this study relate to the booking process and are not exhaustive of all practical barriers to cervical screening. They may not reflect booking processes in other countries.

## INTRODUCTION

Cervical screening programmes are designed to reduce the incidence and mortality rate of cervical cancer.[1] In Great Britain, all eligible women aged 25–64 registered with a general practitioner (GP) are invited to be screened for the presence of abnormal cell changes in the cervix, which could, if undetected and untreated, develop into cervical cancer. The efficacy of the programme has been widely acknowledged;[2] however, the success of any screening programme is dependent on good coverage. In 2017, coverage (ie, the percentage of eligible women recorded as adequately screened) was 72%, well below the national target of 80% and in keeping with a trend of decreasing screening coverage.

Reasons for screening non-attendance are complex and differ depending on socio-demographic factors such as age, socioeconomic status and marital status.[3–6] Emotional barriers including embarrassment, fear of pain and negative experiences are often reported, particularly in qualitative studies.[7–9] While these barriers undoubtedly need to be addressed, practical barriers have been found to be more predictive of screening status than emotional barriers.[10] Recent research showed that over half of women overdue for cervical screening have positive intentions to attend.[11] While this is encouraging, intentions are frequently not translated into action.[12 13]

Weinstein used a 'messy desk' analogy to help explain the problem of translating intentions into action.[14] He proposed that we do not carry out errands in a logical sequence,

**BMJ**

but rather in a haphazard manner, acting on 'to-do' list items when we feel pressure, when items need to be actioned quickly, when prompted or because of personal preference. More recently, Sheeran and Webb identified three key problems (or 'TRIALS') people might encounter when trying to realise their intentions; (i) they fail to get started (eg, forget to act or miss an opportunity to act), (ii) they fail to keep the goal on track (fail to monitor the goal, face competing thoughts or distractions) and (iii) they fail to close (do not quite meet the goal).[15]

Women receive a posted letter inviting them to book a screening appointment. The letter states the recipient 'can make an appointment for cervical screening by phoning *(their)* GP surgery'. GP surgery hours generally coincide with 'normal' working hours, presenting several practical barriers for women who are in full-time employment or who have caring responsibilities, both in terms of phoning and attending a GP surgery. Previous research has identified that many women find the booking process arduous and inflexible.[3]

Few studies have assessed alternative methods of inviting women for cervical screening.[16] The most recent Cochrane review of interventions to improve uptake[16] reported two studies from the 1980s and 1990s, which found that participants who received a telephone invitation were significantly more likely to attend than those who received a letter.[17 18] Studies which have examined the utility of more recent technological developments to invite women are lacking.[19] There is also a paucity of literature concerning alternative booking methods for cervical screening, most likely due to limited booking options being available until recently. One trial investigated the efficacy of online booking among first time invitees.[20] The intervention group booked slightly more appointments within 3 months (2.18% higher than the control group); however, this was not statistically significant.[20] The authors noted that the way the online booking system was offered could account for the lack of support (in a letter, participants were asked to visit a website to book at one of the three sexual health clinics). Hence, other forms of online booking may be desirable to women.

New technologies offer opportunities for editing the architecture of the invitation and booking system in ways that may help to overcome some of the challenges women face between forming a positive intention and translating this into behaviour, as highlighted in the TRIALS model. For example, online booking methods may reduce the likelihood that women would fail to get started, given that opportunities to act (ie, book an appointment) are not limited to GP practice opening hours. The present study explored practical barriers to booking an appointment among two groups: women who are up-to-date with screening ('maintainers') and women who intend to be screened but are currently overdue ('intenders'). Our aim was to examine between-group differences which may account for this intention-behaviour gap among 'intenders'. We also assessed invitation and booking preferences and explored whether these might help to overcome practical barriers.

## METHODS

### Participants

Participants were recruited by Kantar TNS UK as part of their omnibus survey. The TNS omnibus survey recruits a new sample of 2000–4000 men and women living in Great Britain on a weekly basis and asks questions on a range of topics commissioned by external companies. Recruitment uses random location sampling to identify areas for sampling participants using the 2011 Census and the Postcode Address File. Recruiters visit homes in the identified areas and knock on doors asking those who answer to participate. All interviews are conducted in English. Quotas are set at each location for age, gender, working status and presence of children in the household.

Women who were eligible for cervical screening and had not previously been diagnosed with cervical cancer, were asked to report their past attendance at cervical screening and future intention to attend (see online supplement 1). Responses to these questions were used to classify women as 'intenders' (intended to be screened but were currently overdue), 'maintainers' (up-to-date with screening and intending to go in the future) or 'other' (never heard of screening, never been invited and decided not to be screened). A sample of 600 women was expected to allow us to establish a significant difference of 5% between preferred booking options in the two groups of attenders within ±8% with 95% confidence.

### Procedure

Data were collected between April and May 2018. Face-to-face computer-assisted personal interviews were used to collect data. Kantar TNS provided anonymised data to University College London for analysis.

### Measures

*Invitation preferences:* Participants were asked whether several different modes of communication were acceptable to them as a means of being invited to book a cervical screening appointment (see online supplement 1). Participants' responses were recoded as 'acceptable' (if they responded quite acceptable/very acceptable) or 'unacceptable/ambivalent' (if they responded quite unacceptable/very unacceptable/neither unacceptable nor acceptable). Participants who responded quite/very unacceptable were asked to explain why (open response).

*Practical barriers to booking an appointment:* Participants were asked to respond to a list of barriers, which were based on the key problems outlined in the TRIALS model.[15] Statements addressing the key problem of 'failing to get started' included 'It is easy for me to find time to read a letter like this' and 'I might forget to book an appointment after reading this letter'. Statements addressing 'failing to keep the goal on track' included 'It is difficult for me to call my GP practice during their

opening hours' and 'I find it difficult to get through to a receptionist when I phone my GP practice'. Women were then asked to state which booking attributes were important to them, the aim of which was to address factors that might influence 'failure to close' (ie, being able to book the appointment).

*Booking preferences:* Participants were asked to indicate how likely they would be to use different booking methods. The feasibility of these methods were informally discussed with stakeholders from the NHS national screening programme and with representatives from a technology company, who develop methods of improving access to healthcare. Participants' responses were recoded as 'likely to use' a method (if they responded quite likely/very likely) or 'not likely to use/ambivalent' (if they responded quite unlikely/very unlikely/neither unlikely nor likely). Participants were also asked to indicate which booking methods they had used in the past for any GP appointment.

*Sociodemographic and background factors:* Data regarding age, ethnicity, education level, employment status, marital status, social grade, child/carer responsibilities and smartphone ownership were also collected. Social grade is determined by the occupation of the Chief Income Earner in the household and is classified as follows: AB managerial/professional; C1 supervisory; C2 skilled manual; D semi-skilled/unskilled manual; E casual workers/unemployed.[21]

## Patient and public involvement statement

The study was supported by a patient and public involvement (PPI) group who provided input into the contents of the survey. A group of 10 screening-eligible women were invited to guide and refine the survey questions. Women who were both up-to-date and overdue were represented in the group. The group helped to establish the perceived difficulty of the questions (eg, unknown terms, ambiguous concepts, long and overly complex questions) and omissions from the survey. The questions and response options were tailored based on feedback provided by this PPI group.

## Analyses

All analyses were conducted using IBM SPSS V.22. $\chi^2$ analyses were conducted to test for significant differences in participant demographics between 'intenders' and 'maintainers'. Descriptive statistics were conducted to assess booking history and smartphone/mobile phone ownership across all participants. For each of the six practical barrier statements, any positively framed items were reverse-scored so that a higher score was indicative of a barrier for all items. Total practical barrier scores were created by allocating a score of 1 for each barrier statement that a participant 'agreed' or 'strongly agreed' with and adding these together (possible range 0–6). Independent samples t-tests were conducted to assess differences in the mean barriers scores between 'intenders' and 'maintainers'. A series of binary logistic regressions were then conducted to assess the associations

between endorsing each barrier/booking attribute and the unadjusted odds for being an 'intender' (vs a 'maintainer'). A series of univariable logistic regressions were conducted to explore whether sociodemographic factors, screening status and number of practical barriers reported were associated with invitation (acceptable vs unacceptable/ambivalent) and booking preferences (likely to use vs unlikely to use/ambivalent). Participants responding do not know or not applicable were excluded. Multivariable logistic regressions are presented as online supplement 2.

## RESULTS

### Sample characteristics

2509 eligible respondents (ie, women aged 25–64 years) completed the Kantar TNS survey. After exclusions, 1548 (78%) were up-to-date and 445 (22%) were overdue for screening. Our questions on invitation and booking preferences for cervical screening were asked to all women who were classified as 'intenders' and women who were classified as 'maintainers' in week 1. See online supplement 3 for survey inclusion flow diagram.

Sample characteristics for participants classified as 'intenders' (n=255) and 'maintainers' (n=359) are presented in table 1. Mean age was 41.69 years (SD=10.84, range: 25–64 years), the majority self-identified as White (89%), were employed (64%), married or cohabiting (67%) and had regular caring responsibilities (ie, for children/parents; 63%). 'Intenders' (mean=39.41; SD=9.94) were significantly younger than 'maintainers' (mean=43.31; SD=11.16); t(612)=4.47, p<0.001.

The majority of women had previously booked by phoning the practice (89%), over one-third had booked in person (39%) and 14% had booked on a website. 'Maintainers' were significantly more likely to have previously booked on a website than 'intenders' (see table 1). The majority of participants had a smartphone (87%), fewer women had a mobile phone which was not a smartphone (11%) and a small minority had no mobile phone (2%).

### Practical barriers to appointment booking and desired attributes

Over two-thirds of women reported one or more barriers to booking (69%); mean number of reported barriers was 1.21 (SD=1.06). 'Intenders' (mean=1.36; SD=1.06) reported slightly more barriers than 'maintainers' overall (mean=1.10; SD=1.04; t(612)=3.03, p<0.01). The most commonly endorsed barrier was 'I find it difficult to get through to a receptionist when I phone my GP practice' (50% of participants 'strongly agreed' or 'agreed'), followed by 'It is difficult for me to call my GP practice during their opening hours' (31%) and 'I might forget to book an appointment after reading this letter' (31%). Practical barriers to appointment booking and booking characteristics considered to be important are outlined in table 2. The 'intenders' group were significantly more likely to endorse the statement 'I might forget to book an appointment after reading this letter' than 'maintainers'. 'Intenders' were also more likely

**Table 1** Sample characteristics (n=614)

| | Overall (n=614) N (%) | Maintainers (n=359) N (%) | Intenders (n=255) N (%) | Difference between maintainers and intenders $\chi^2$ (df), P value |
|---|---|---|---|---|
| Age (years) | | | | 14.16 (3), <0.001 |
| 25–34 | 192 (31.3) | 103 (28.7) | 89 (34.9) | |
| 35–44 | 183 (29.8) | 95 (26.5) | 88 (34.5) | |
| 45–54 | 137 (22.3) | 88 (24.5) | 49 (19.2) | |
| 55–64 | 102 (16.6) | 73 (20.3) | 29 (11.4) | |
| Ethnicity | | | | 0.10 (1), 0.76 |
| Any white | 547 (89.1) | 321 (89.4) | 226 (88.6) | |
| All other groups | 67 (10.9) | 38 (10.6) | 29 (11.4) | |
| Education level | | | | 2.12 (4), 0.71 |
| GCSE or below | 180 (29.3) | 108 (30.1) | 72 (28.2) | |
| A level or equivalent | 71 (11.6) | 45 (12.5) | 26 (10.2) | |
| College qualification | 115 (18.7) | 62 (17.3) | 53 (20.8) | |
| Degree or higher | 213 (34.7) | 125 (34.8) | 88 (34.5) | |
| Other | 35 (5.7) | 19 (5.3) | 16 (6.3) | |
| Employment status | | | | 3.19 (2), 0.20 |
| Employed (full-time/part-time) | 392 (63.8) | 234 (65.2) | 158 (62.0) | |
| Unemployed | 182 (29.6) | 98 (27.3) | 84 (32.9) | |
| Other (studying/retired) | 40 (6.5) | 27 (7.5) | 13 (5.1) | |
| Marital status | | | | 2.89 (2), 0.24 |
| Single | 129 (21.0) | 67 (18.7) | 62 (24.3) | |
| Married/living as married | 413 (67.3) | 249 (69.4) | 164 (64.3) | |
| Widowed/divorced/separated | 72 (11.7) | 43 (12.0) | 29 (11.4) | |
| Parent/carer role | | | | 0.62 (0.45), 0.43 |
| Yes | 387 (63.0) | 221 (61.6) | 166 (65.1) | |
| No | 222 (36.2) | 134 (37.3) | 88 (34.5) | |
| Social status | | | | 7.93 (4), 0.09 |
| AB (highest) | 134 (21.8) | 90 (25.1) | 44 (17.3) | |
| C1 | 157 (25.6) | 88 (24.5) | 69 (27.1) | |
| C2 | 142 (23.1) | 84 (23.4) | 58 (22.7) | |
| D | 93 (15.1) | 54 (15.0) | 39 (15.3) | |
| E (lowest) | 88 (14.3) | 43 (12.0) | 45 (17.6) | |
| Booking history (yes/no) | | | | |
| Phoned the practice | 545 (88.8) | 316 (88.0) | 229 (89.8) | 0.47 (1), 0.49 |
| At reception (in person) | 240 (39.1) | 145 (40.4) | 95 (37.3) | 0.62 (1), 0.43 |
| 24 hours automated service | 23 (3.7) | 14 (3.9) | 9 (3.5) | 0.06 (1), 0.81 |
| Text-message | 7 (1.1) | 4 (1.1) | 3 (1.2) | 0.01 (1), 0.94 |
| Website | 85 (13.8) | 60 (16.7) | 25 (9.8) | 5.97 (1), <0.05 |
| Smartphone app | 23 (3.7) | 15 (4.2) | 8 (3.1) | 0.45 (1), 0.50 |
| Phone ownership | | | | 0.72 (2), 0.70 |
| Smartphone | 533 (86.8) | 315 (87.7) | 218 (85.5) | |
| Non-smartphone mobile | 67 (10.9) | 36 (10.0) | 31 (12.2) | |
| No phone | 14 (2.3) | 8 (2.2) | 6 (2.4) | |

GP, general practitioner.

| | All (n=614) | 'Maintainers' (n=359) | 'Intenders' (n=255) | OR for being an 'intender' (95% CI) |
|---|---|---|---|---|
| | N (%) | N (%) | N (%) | |
| **Practical barriers to booking screening (% agree/strongly agree)** | | | | |
| It is (*not*) easy for me to find time to read a letter like this | 25 (4.1) | 15 (4.2) | 10 (3.9) | 0.94 (0.41 to 2.12) |
| I might forget to book an appointment after reading this letter | 187 (30.5) | 76 (21.2) | 111 (43.5) | 2.87 (2.01 to 4.09)** |
| It is difficult for me to call my GP practice during their opening hours | 192 (31.3) | 108 (30.1) | 84 (32.9) | 1.14 (0.81 to 1.61) |
| I (*do not*) have access to a telephone/mobile with phone credit/minutes to call my GP practice | 13 (2.1) | 8 (2.2) | 5 (2.0) | 0.88 (0.28 to 2.71) |
| I would (*not*) find it easy to find the phone number for my GP practice to contact them | 19 (3.1) | 11 (3.1) | 8 (3.1) | 1.01 (0.41 to 2.59) |
| I find it difficult to get through to a receptionist when I phone my GP practice | 306 (49.8) | 177 (49.3) | 129 (50.6) | 1.05 (0.76 to 1.45) |
| **Booking attributes (% saying quite/very important)** | | | | |
| Ease of booking | 519 (84.5) | 305 (85.0) | 214 (83.9) | 0.92 (0.59 to 1.44) |
| Choice of appointments | 486 (79.2) | 280 (78.0) | 206 (80.8) | 1.19 (0.83 to 1.77) |
| Being able to change an appointment after booking | 474 (77.2) | 274 (76.3) | 200 (78.4) | 1.13 (0.77 to 1.66) |
| How long it takes to book appointment | 424 (69.1) | 235 (65.5) | 189 (74.1) | 1.51 (1.06 to 2.15)* |
| Waiting time for next available appointment | 428 (69.7) | 245 (68.2) | 183 (71.8) | 1.18 (0.83 to 1.68) |
| Privacy when booking appointment | 410 (66.8) | 230 (64.1) | 180 (70.6) | 1.35 (0.95 to 1.90) |
| Being able to talk with a healthcare professional when booking | 345 (56.2) | 195 (54.3) | 150 (58.8) | 1.20 (0.87 to 1.66) |
| Being able to book an appointment when the GP practice is shut | 284 (46.3) | 173 (48.2) | 111 (43.5) | 0.83 (0.60 to 1.15)† |
| Cost of making booking (ie, phone credit) | 166 (27.0) | 94 (26.2) | 72 (28.2) | 1.11 (0.77 to 1.59) |

*p<0.05.
**p<0.001.
†30% missing data for this variable.
GP, general practitioner.

to state 'How long it takes to book the appointment' was important to them than 'maintainers'.

### Invitation preferences

Posted letters emerged as the most acceptable invitation mode followed by text-messages (see table 3). Sociodemographic predictors of the acceptability of each modality are shown in table 3. Text-message, email and mobile call invitations were less acceptable to women aged 55–64; these associations remained significant in multivariable analyses (see online supplement 2). Mobile and landline call invites were more acceptable to women from lower socioeconomic backgrounds and this remained significant in multivariable analyses for mobile invites. Reasons for considering invitation modes as unacceptable are provided in online supplement 4; fears about missing a phone call/email or text and privacy concerns were commonly cited. Many participants also reported they had no landline phone.

### Phone-based booking preferences

Most women said they were likely to book by phoning their GP practice (90%; see table 4). Older women were significantly less likely to say they would call a 24 hours automated service than women aged 25–34 (44% vs 63%). Women with caring responsibilities were more likely to say they would request a call-back compared with women with no caring responsibilities (63% vs 51%). 'Maintainers' were less likely to say they would request a call-back than 'intenders' (54% vs 66%). These associations remained significant in multivariable analyses. Women who cited three or more barriers were more likely to say they would call a 24 hours automated service but this association was not significant in multivariable analyses.

### Online booking preferences

Booking on a website using a smartphone (59%) was the preferred online booking method (see table 5). Older women (55–64 years) were less likely to say they would book online than younger women (25–34 years). Women in lower social grades were less likely than women in the highest grade to state they would book on a website, either using a desktop or smartphone. Participants who were studying or retired were less likely than those employed to say they would book online (either on a website using a smartphone: 41% vs 64%,

**Table 3** Univariable logistic regression models of predictors of the acceptability of cervical screening invitation modalities

| | Posted letter (n=598) | | Text-message (n=597) | | Email (n=592) | | Mobile phone call (n=598) | | Landline phone call (n=576) | |
|---|---|---|---|---|---|---|---|---|---|---|
| | % | OR (95% CI) | % | OR (95% CI) | % | OR (95% CI) | % | OR (95% CI) | % | OR (95% CI) |
| All participants | 92.5 | 1.00 | 80.7 | 1.00 | 75.2 | 1.00 | 75.8 | 1.00 | 62.3 | 1.00 |
| **Age group** | | | | | | | | | | |
| 25–34 | 94.7 | 1.00 | 86.7 | 1.00 | 80.9 | 1.00 | 82.4 | 1.00 | 65.0 | 1.00 |
| 35–44 | 92.1 | 0.66 (0.28 to 1.52) | 84.2 | 0.82 (0.46 to 1.46) | 78.2 | 0.85 (0.51 to 1.41) | 80.8 | 0.90 (0.53 to 1.52) | 67.5 | 1.12 (0.72 to 1.74) |
| 45–54 | 87.5 | 0.40 (0.18 to 0.89)* | 78.7 | 0.57 (0.31 to 1.02) | 74.1 | 0.68 (0.40 to 1.15) | 69.1 | 0.48 (0.28 to 0.80)* | 53.4 | 0.62 (0.39 to 0.98)* |
| 55–64 | 95.9 | 1.33 (0.41 to 4.35) | 65.6 | 0.29 (0.16 to 0.53)*** | 60.0 | 0.36 (0.21 to 0.62)*** | 62.9 | 0.36 (0.21 to 0.63)*** | 60.4 | 0.82 (0.49 to 1.37) |
| **Social grade** | | | | | | | | | | |
| AB | 91.6 | 1.00 | 77.9 | 1.00 | 81.3 | 1.00 | 64.6 | 1.00 | 51.6 | 1.00 |
| C1 | 91.2 | 0.95 (0.41 to 2.19) | 81.8 | 1.27 (0.71 to 2.29) | 78.2 | 0.83 (0.46 to 1.50) | 71.6 | 1.38 (0.83 to 2.29) | 57.0 | 1.25 (0.77 to 2.01) |
| C2 | 97.2 | 3.14 (0.97 to 10.12) | 79.4 | 1.10 (0.62 to 1.96) | 73.0 | 0.63 (0.35 to 1.12) | 82.4 | 2.56 (1.46 to 4.50)** | 67.9 | 1.99 (1.21 to 3.27)** |
| D | 95.6 | 1.99 (0.61 to 6.47) | 86.8 | 1.87 (0.90 to 3.90) | 79.1 | 0.87 (0.45 to 1.71) | 79.1 | 2.08 (1.12 to 3.86)* | 67.8 | 1.98 (1.12 to 3.49)* |
| E | 85.2 | 0.53 (0.23 to 1.24) | 79.1 | 1.07 (0.55 to 2.09) | 60.0 | 0.35 (0.19 to 0.64)* | 85.1 | 3.12 (1.56 to 6.22)** | 73.2 | 2.56 (1.41 to 4.66)** |
| **Employment** | | | | | | | | | | |
| Employed | 93.0 | 1.00 | 80.3 | 1.00 | 77.8 | 1.00 | 73.6 | 1.00 | 58.2 | 1.00 |
| Unemployed | 89.8 | 0.66 (0.36 to 1.24) | 82.9 | 1.19 (0.75 to 1.90) | 70.5 | 0.68 (0.46 to 1.02) | 84.0 | 1.89 (1.19 to 3.00)** | 72.0 | 1.84 (1.25 to 2.74)** |
| Other (studying/retired) | 100.0 | – | 75.7 | 0.77 (0.35 to 1.69) | 69.4 | 0.65 (0.31 to 1.37) | 59.5 | 0.53 (0.26 to 1.06) | 59.5 | 1.05 (0.53 to 2.09) |
| **Ethnicity** | | | | | | | | | | |
| Any white | 93.3 | 1.00 | 79.6 | 1.00 | 73.5 | 1.00 | 74.9 | 1.00 | 61.4 | 1.00 |
| All other groups | 85.9 | 0.44 (0.20 to 0.97) | 90.5 | 2.44 (1.02 to 5.80)* | 88.9 | 2.88 (1.28 to 6.47)** | 82.8 | 1.61 (0.82 to 3.18) | 69.8 | 1.46 (0.83 to 2.57) |
| **Caring responsibilities** | | | | | | | | | | |
| No | 91.7 | 1.00 | 78.0 | 1.00 | 71.8 | 1.00 | 68.1 | 1.00 | 58.7 | 1.00 |
| Yes | 92.9 | 1.19 (0.64 to 2.22) | 82.2 | 1.30 (0.86 to 1.97) | 77.0 | 1.31 (0.90 to 1.92) | 80.1 | 1.88 (1.29 to 2.76)** | 64.3 | 1.27 (0.90 to 1.80) |
| **Screening status** | | | | | | | | | | |
| Intender | 91.1 | 1.00 | 82.2 | 1.00 | 75.1 | 1.00 | 76.5 | 1.00 | 61.9 | 1.00 |
| Maintainer | 93.4 | 1.38 (0.75 to 2.54) | 79.7 | 0.85 (0.56 to 1.29) | 75.2 | 1.01 (0.69 to 1.47) | 75.2 | 0.93 (0.64 to 1.36) | 62.6 | 1.03 (0.73 to 1.45) |
| **Practical barriers** | | | | | | | | | | |
| 0 barriers | 94.0 | 1.00 | 81.0 | 1.00 | 73.1 | 1.00 | 77.6 | 1.00 | 60.7 | 1.00 |
| 1 barrier | 94.1 | 1.02 (0.43 to 2.42) | 79.1 | 0.89 (0.54 to 1.48) | 73.7 | 1.03 (0.64 to 1.64) | 75.9 | 0.91 (0.56 to 1.48) | 60.4 | 0.99 (0.65 to 1.51) |
| 2 barriers | 92.5 | 0.79 (0.33 to 1.87) | 81.6 | 1.04 (0.60 to 1.82) | 77.2 | 1.25 (0.75 to 2.08) | 73.6 | 0.81 (0.49 to 1.34) | 65.0 | 1.20 (0.76 to 1.91) |
| 3 or more barriers | 84.8 | 0.36 (0.15 to 0.84)* | 82.3 | 1.09 (0.55 to 2.16) | 79.7 | 1.45 (0.77 to 2.75) | 75.0 | 0.87 (0.47 to 1.60) | 65.8 | 1.25 (0.71 to 2.19) |

*P<0.05.
**P<0.01.
***P<0.05.
Reference group: 'unacceptable /ambivalent'.

**Table 4** Univariable logistic regression models of predictors of phone-based booking preferences

| | Calling the GP (n=596) | | Calling a 24 hours automated service (n=590) | | Requesting a call-back (n=593) | |
|---|---|---|---|---|---|---|
| | % likely to book by… | OR (95% CI) | % likely to book by… | OR (95% CI) | % likely to book by… | OR (95% CI) |
| All participants | 92.3 | | 53.7 | | 59.0 | |
| Age group | | | | | | |
| 25–34 | 93.0 | 1.00 | 63.2 | 1.00 | 61.0 | 1.00 |
| 35–44 | 92.7 | 0.94 (0.42 to 2.09) | 54.8 | 0.71 (0.46 to 1.07) | 64.2 | 1.15 (0.75 to 1.76) |
| 45–54 | 89.7 | 0.65 (0.30 to 1.43) | 45.9 | 0.49 (0.31 to 0.78)** | 48.9 | 0.61 (0.39 to 0.96)* |
| 55–64 | 93.8 | 1.12 (0.41 to 3.05) | 44.2 | 0.46 (0.28 to 0.76)** | 60.0 | 0.96 (0.58 to 1.59) |
| Social grade | | | | | | |
| AB | 91.5 | 1.00 | 51.5 | 1.00 | 55.4 | 1.00 |
| C1 | 91.8 | 1.04 (0.44 to 2.44) | 53.1 | 1.07 (0.33 to 1.71) | 52.7 | 0.90 (0.56 to 1.45) |
| C2 | 93.6 | 1.36 (0.54 to 3.39) | 58.9 | 1.35 (0.83 to 2.18) | 60.3 | 1.22 (0.75 to 1.98) |
| D | 94.5 | 1.59 (0.53 to 4.74) | 54.4 | 1.12 (0.66 to 1.93) | 65.6 | 1.53 (0.88 to 2.67) |
| E | 89.7 | 0.80 (0.32 to 2.02) | 48.8 | 0.90 (0.52 to 1.55) | 66.3 | 1.58 (0.90 to 2.79) |
| Employment | | | | | | |
| Employed | 91.7 | 1.00 | 51.3 | 1.00 | 57.3 | 1.00 |
| Unemployed | 92.6 | 1.14 (0.58 to 2.23) | 52.3 | 0.92 (0.64 to 1.32) | 63.6 | 1.30 (0.91 to 1.88) |
| Other (studying/ retired) | 97.2 | 3.18 (0.42 to 23.99) | 54.3 | 1.00 (0.50 to 2.00) | 54.3 | 0.88 (0.44 to 1.77) |
| Ethnicity | | | | | | |
| Any white | 92.3 | 1.00 | 52.3 | 1.00 | 58.2 | 1.00 |
| All other groups | 92.2 | 0.99 (0.38 to 2.59) | 65.6 | 1.74 (1.01 to 3.00) | 65.6 | 1.37 (0.80 to 2.36) |
| Caring responsibilities | | | | | | |
| No | 93 | 1.00 | 53.3 | 1.00 | 51.4 | 1.00 |
| Yes | 91.9 | 0.85 (0.45 to 1.62) | 54.0 | 1.03 (0.73 to 1.44)* | 63.3 | 1.63 (1.16 to 2.29)** |
| Screening status | | | | | | |
| Intender | 91.1 | 1.00 | 56.1 | 1.00 | 65.7 | 1.00 |
| Maintainer | 93.1 | 1.33 (0.73 to 2.44) | 52.0 | 0.85 (0.61 to 1.18) | 54.3 | 0.62 (0.44 to 0.90)** |
| Practical barriers | | | | | | |
| 0 barriers | 93.4 | 1.00 | 50.9 | 1.00 | 57.2 | 1.00 |
| 1 barrier | 93.6 | 1.04 (0.46 to 2.38) | 48.4 | 0.92 (0.61 to 1.38) | 53.7 | 0.87 (0.58 to 1.31) |
| 2 barriers | 93.9 | 1.09 (0.45 to 2.66) | 59.0 | 1.41 (0.91 to 2.19) | 64.8 | 1.38 (0.88 to 2.16) |
| 3 or more barriers cited | 83.8 | 0.37 (0.16 to 0.84)* | 64.1 | 1.75 (1.01 to 3.02)* | 65.0 | 1.39 (0.80 to 2.40) |

*P<0.05.
**P<0.01.
***P<0.001.
Reference group: 'not likely to use/ambivalent'.
GP, general practitioner.

or through an app: 22% vs 54%). Women who reported two or more barriers were more likely to report that they would use all online booking methods compared with women who reported no barriers (see table 5). Age, social grade, employment status and number of barriers remained significant in multivariable analyses.

## DISCUSSION

This study examined women's practical barriers to booking a cervical screening appointment and assessed whether invitation and booking preferences are associated with reported barriers, sociodemographic factors and screening status. Approximately one-third of all

**Table 5** Univariable logistic regression models of predictors of online booking preferences

| | Booking on a website using a desktop/laptop (n=589) | | Booking on a website using a smartphone† (n=513) | | Downloading an app to your smartphone† (n=517) | |
|---|---|---|---|---|---|---|
| | % likely to book by… | OR (95% CI) | % likely to book by… | OR (95% CI) | % likely to book by… | OR (95% CI) |
| All participants | 60.3 | | 58.8 | | 49.1 | |
| Age group | | | | | | |
| 25–34 | 71.0 | 1.00 | 74.5 | 1.00 | 67.6 | 1.00 |
| 35–44 | 61.9 | 0.66 (0.43 to 1.03) | 64.8 | 0.63 (0.40 to 0.99)* | 53.7 | 0.56 (0.36 to 0.85)** |
| 45–54 | 55.2 | 0.50 (0.32 to 0.80)** | 47.0 | 0.30 (0.19 to 0.49)*** | 36.3 | 0.27 (0.17 to 0.44)*** |
| 55–64 | 43.8 | 0.32 (0.19 to 0.53)*** | 34.0 | 0.18 (0.10 to 0.30)*** | 22.9 | 0.14 (0.08 to 0.25)*** |
| Social grade | | | | | | |
| AB | 72.3 | 1.00 | 70.0 | 1.00 | 53.1 | 1.00 |
| C1 | 61.1 | 0.60 (0.36 to 1.00) | 63.9 | 0.76 (0.46 to 1.26) | 53.4 | 1.01 (0.63 to 1.63) |
| C2 | 59.3 | 0.56 (0.34 to 0.93)* | 54.3 | 0.51 (0.31 to 0.84)** | 48.9 | 0.85 (0.53 to 1.37) |
| D | 58.2 | 0.53 (0.30 to 0.94)* | 54.9 | 0.52 (0.30 to 0.91)* | 47.3 | 0.79 (0.46 to 1.36) |
| E | 44.0 | 0.30 (0.17 to 0.54)*** | 44.7 | 0.35 (0.20 to 0.61)*** | 36.6 | 0.53 (0.31 to 0.93)* |
| Employment | | | | | | |
| Employed | 64.5 | 1.00 | 63.7 | 1.00 | 53.5 | 1.00 |
| Unemployed | 52.6 | 0.61 (0.43 to 0.88)** | 51.7 | 0.61 (0.43 to 0.88)** | 44.8 | 0.71 (0.49 to 1.01) |
| Other (studying/ retired) | 52.8 | 0.62 (0.31 to 1.22) | 41.2 | 0.40 (0.20 to 0.82)* | 22.2 | 0.25 (0.11 to 0.56)** |
| Ethnicity | | | | | | |
| Any white | 59.7 | 1.00 | 57.7 | 1.00 | 48.4 | 1.00 |
| All other groups | 65.1 | 1.26 (0.73 to 2.17) | 68.3 | 1.58 (0.90 to 2.75) | 54.7 | 1.29 (0.77 to 2.17) |
| Caring responsibilities | | | | | | |
| No | 60.6 | 1.00 | 54.2 | 1.00 | 42.5 | 1.00 |
| Yes | 60.1 | 0.98 (0.70 to 1.38) | 61.4 | 1.34 (0.96 to 1.89) | 52.8 | 1.51 (1.08 to 2.12)* |
| Screening status | | | | | | |
| Intender | 59.6 | 1.00 | 59.2 | 1.00 | 52.8 | 1.00 |
| Maintainer | 60.8 | 1.05 (0.75 to 1.47) | 58.6 | 0.98 (0.70 to 1.36) | 46.4 | 0.77 (0.56 to 1.07) |
| Practical barriers | | | | | | |
| 0 barriers | 50.8 | 1.00 | 48.9 | 1.00 | 39 | 1.00 |
| 1 barrier | 60.0 | 1.45 (0.96 to 2.20) | 55.4 | 1.30 (0.86 to 1.96) | 45.2 | 1.29 (0.85 to 1.95) |
| 2 barriers | 67.4 | 2.00 (1.27 to 3.14)** | 68.1 | 2.23 (1.41 to 3.52)** | 58.3 | 2.19 (1.40 to 3.42)** |
| 3 or more barriers | 69.6 | 2.22 (1.27 to 3.89)** | 73.1 | 2.84 (1.59 to 5.07)*** | 64.6 | 2.85 (1.65 to 4.93)*** |

*P<0.05.
**P<0.01.
***P<0.001.
†Participants with no smartphone removed from analyses (n=81).
Reference group: 'not likely to use/ambivalent'.
GP, general practitioner.

women reported that it is difficult to phone their GP practice within opening hours and half reported that it is difficult to get through to a receptionist. Although the survey found that 'intenders' experience slightly more practical barriers to screening than 'maintainers', endorsement of barriers across the sample suggests that both groups need more support in booking an appointment.

'Intenders' were more likely to report that they would forget to book an appointment after reading the screening letter than 'maintainers'. This key problem relates to a 'failure to get started', which is a first barrier people face

between forming an intention and translating this into behaviour.[15] Written reminders are an integral part of the screening programme and there is good evidence to show these improve uptake,[16] but in their current format, these reminders do not seem to help all women to remember to book their appointment. Future research might explore methods of increasing the salience of cervical screening among invitees (eg, employing implementation intentions).[22] The use of text-message reminders has shown promise in other screening contexts.[23] 'Intenders' were also more likely to say that the length of time needed

to book an appointment was important to them. Since all women eligible for cervical screening fall within the working age population, and GP opening hours generally overlap with working hours, it is likely that this cohort face competing obligations,[24] and, as a result 'fail to keep their goal on track'.[15] The rate of female employment (16–64 years) has increased from 62.2% in 1994, when coverage was high (85%; 5 yearly coverage for women aged 20–64)[25] to 70.5% in 2017.[26] Alternative booking methods may provide more flexibility.

Women who reported more barriers showed greater interest in using alternative booking methods. Specifically, participants who reported two or more barriers were more likely to say that they would book on a website or through an app. This is perhaps not surprising since these methods overcome the most common practical barriers highlighted by participants, including, difficulty getting through to a receptionist and difficulty calling the practice during opening hours; hence, they 'fail to close'. Nevertheless, while 24 hours automated services offer these same advantages, consistent with previous national surveys,[27] fewer women reported that they would use this booking option. Online booking services are already set up in the majority of GP practices across England for GP appointments; however, a national survey found that over 40% of patients are unaware if there are online booking services at their GP practice.[28] Hence, signposting online booking services, if available for nurse appointments, to groups of the screening-eligible population (ie, younger women who are more likely to be 'intenders') may be an effective means of increasing uptake. This survey suggests that there are likely to be age and socioeconomic inequalities in the use of online bookings. For example, women aged 45–54 years and women aged 55–64 showed less interest in using online booking methods. Thus, ensuring that traditional telephone booking options remains available is important.

Previous research has found that it is very difficult for individuals to maintain intentions even after very brief periods of time (less than 1 min), especially in circumstances where there are competing tasks.[29] Unlike posted letters, which may not be read until the end of the day, text-messages can be delivered at a time when GP practices are open, so women can act immediately on their intentions to book an appointment. Given that text-message invites were considered acceptable to the majority of women across all sociodemographic backgrounds, and have previously been found to be effective in increasing uptake for other national screening programmes,[23] the use of text-message invitations may be a worthwhile intervention to explore. Text-messages within the cervical screening programme have, thus far, been introduced as a booking reminder, rather than as a standalone invitation, which the current study did not specify. Some participants shared concerns that they may miss the message; outlining that text-messages described as a supplemental invitation may have further increased acceptability within the sample. Further research is needed to explore methods of overcoming privacy concerns associated with text-messages, which some of the participants raised.

This study had some limitations. We were unable to collect data on women who elected not to participate in the study. Hence, the response rate and differences between respondents and non-respondents could not be determined. Women in the survey tended to be slightly less deprived and were less likely to be from ethnic minority backgrounds than the population represented in the most recent Census.[30] This suggests that there was a slight bias in participation. This survey was also conducted in English and therefore non-English speakers were not represented. Given ethnic disparities in screening attendance in England,[31] more work is needed to explore methods of overcoming practical barriers to screening for ethnic minority women.

Participation in screening was self-reported. Previous research has found that women tend to over-report their participation in cervical screening programmes,[32 33] thus some of the women classified as 'maintainers' may actually be overdue for screening. Furthermore, although this study explored practical barriers to appointment-booking based on the TRIALS model,[15] several other practical barriers were not assessed. For example, previous research has found that 'intenders' are more likely to have children under the age of 5[11]; childcare may be an additional practical barrier to screening. Thus, the barriers cited in this study are not exhaustive of all practical barriers to screening for women. In addition, the study was designed to reflect the current booking process for cervical screening in Great Britain. While there may be parallels with other countries that have call-recall programmes with paper-based invitations and self-booked appointments in primary care, the findings may not be generalisable to screening programmes in other countries, where the invitation and booking approach differs.

Nevertheless, this was the first study to assess preferences for booking a screening appointment in Great Britain, an important first step in the development of trialling and implementing any of these changes. The invitation and booking process was broken down to identify barriers at each stage and associated preferences which may help women to overcome such barriers. The lack of differences by screening status suggests that changing the architecture should not deter 'maintainers' from participation. Future interventions may assess the efficacy of i) signposting invitees to online booking services, ii) text-messages which are delivered during GP opening hours and iii) sending reminders to reduce the likelihood of forgetting to book an appointment. Implementation research will further determine how best to introduce such changes to the screening infrastructure.

**Contributors** MR (conceptualisation; data analysis; project administration; writing—original draft; writing—review and editing). JW (conceptualisation; supervision; writing—review and editing). LM (conceptualisation; data analysis; supervision; writing—review and editing). All authors approved the final manuscript as submitted.

**Funding**  MR, JW and LM are supported by a Cancer Research UK Career Development Fellowship awarded to JW (C7492/A17219).

**Competing interests**  None declared.

**Patient consent for publication**  Not required.

**Ethics approval**  Ethical approval was granted by University College London Research Ethics Committee (reference: 10353/003).

**Provenance and peer review**  Not commissioned; externally peer reviewed.

**Data sharing statement**  Data used and analysed in the study are available from the corresponding author on request (l.marlow@ucl.ac.uk).

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
