## [Reviewer comments · BMJ Open]

ARTICLE DETAILS

TITLE (PROVISIONAL)	Could changing invitation and booking processes help women translate their cervical screening intentions into action? A population-based survey of women's preferences in Great Britain.
AUTHORS	Ryan, Mairead; Waller, Jo; Marlow, Laura

VERSION 1 – REVIEW

REVIEWER	Margaret Cruickshank University of Aberdeen UK
REVIEW RETURNED	22-Dec-2018

GENERAL COMMENTS	This is a very important issue for all screening programmes - communication/engagement with the eligible population. Current programmes all depend on 19th century technology of the postal service and yet there are many groups within the population who are eligible for screening but have never used the postal service and use other means of communication. The references are relevant and update and make the point clearly from the STRATEGIC trial that although internet booking was used (and should some improvement in screening uptake in younger women at first invitation), the participants were informed of this service by post. We have undertaken qualitative research in this area and found similar issues and concerns regarding alternative means of communication. The authors have used a different and interesting approach to access a wide range of women not just those accessing healthcare. They describe the limitations clearly but have still ended up with a demographic which tends to be more engaged in terms of age, education and deprivation. A means of engaging with hard to reach groups is outwith the scope of this work and it still delivers an important message on communication. Some of the concerns about confidentiality seem to ignore that a letter lying in a communal stairway is also easily accessible and safe delivery of mail cannot be guaranteed. The authors discuss the proportion of GP practices in England which allow internet booking. Is this available in other countries and how generalizable are these results in terms of communication used in screening programmes and uptake of screening?
--

REVIEWER	Alastair Gray Health Economics Research Centre, University of Oxford, UK
REVIEW RETURNED	07-Jan-2019

GENERAL COMMENTS	This paper reports the results of a population survey which asks a series of questions about whether women have participated in cervical cancer screening, their perceived obstacles to participating, and their preferences concerning invitation and booking arrangements. 1) The survey was conducted as part of a national omnibus study. I didn't see much discussion of whether those surveyed are representative nationally of that age/gender group, in terms of employment, socio-economic status, ethnicity etc. 2) A limitation of the study design (acknowledged) is that the non-response/refusal rate is not known. Another (not acknowledged) is that actual participation in the screening programme is self-reported only, and may not be completely accurate. It would be useful to cite any literature comparing self-reported and actual participation. 3) The authors do not report in tabular form some of the key responses to the questions they used. For example, Have you ever been diagnosed with cervical cancer, Have you had cervical cancer screening 1-7 responses, will you go when next invited 1-4 responses. Also the number of barriers is mentioned in summary but not tabulated in detail. A couple of summary remarks are made about these in the Results but much more detail could be given and discussed. 4) How are Intenders actually defined? The text says "intended to be screened but were currently overdue". How was this constructed from the available questions? Are the yes probably and yes definitely grouped together? What are the proportions of these? 5) The way other responses are dichotomised is described in a confusing way: for example, if a mode of communication is deemed acceptable: it seems to imply that acceptable includes quite or very acceptable and unacceptable includes quite/very unacceptable but also neither unacceptable or acceptable, don't know or not applicable. Same with Likelihood of using a method of booking. Is this just poorly described? And how many actually were in the neither one nor another, or don't know categories. And what does not applicable mean, if these are hypothetical questions about acceptability of different things? 6) All results are presented in univariate tables. It would be worth conducting and reporting a multivariate analysis as some of these variables are likely to be strongly correlated. 7) The statement in the last paragraph of the discussion that because preferences did not seem to vary much by screening status, this negates ANY concern that changing the current screening programme might deter current participants, seems very strong and should at least be qualified. 8) PPI statement: should this say both up-to-date and overdue?
--

REVIEWER	Jasmin Tiro University of Texas Southwestern, United States
REVIEW RETURNED	17-Jan-2019

GENERAL COMMENTS	This study surveyed British women to understand practical barriers to booking a cervical cancer screening appointment and identify preferred booking modes. This was a novel exploration of the
---

	intention-behavior gap and investigators used the TRIALS framework to identify which problems most interfere with the desired action (booking an appointment). Other strengths include the representative sample, comprehensive measurement of practical barriers, and stratified analysis by past behavior (maintainers vs. intenders). The main weakness of the paper was whether the survey was informed by implementation science and engaged the perspectives of general practice and National Health Service stakeholders. It was unclear whether all invitation and booking options are feasible to implement and should have been offered to women. Specific questions and suggestions are noted below.  1. Abstract, conclusion: what do you mean by tailoring? Will you ask GPs to note each woman's preference and then match the appointment booking process to their preferred method? Or are you just trying to determine which options should be offered to women? 2. Introduction: can you say more specifically what attributes of the alternative booking methods address problems highlighted by Sheeran/Webb's TRIALS framework? Similarly, can you interpret your findings with those attributes in mind? For example, were you surprised that more women didn't prefer the 24 hour automated service or call back options. 3. Methods: did you engage GP and NHS stakeholders in which alternative booking methods are feasible to implement (e.g., have resources, capacity)? If not, then this needs to be acknowledged in the Discussion and future studies should explore these implementation science issues or consider hybrid-effectiveness designs when testing proposed interventions (e.g., text messages during open GP hours). 4. Discussion: Besides making it easier to quickly act in booking an appointment, what are other behavioral determinants that could be addressed to respond to "forgetting to book" (failing to start)? Should investigators try to enhance urgency or motivation?
--	--

VERSION 1 – AUTHOR RESPONSE

Reviewers' Comments to Author:

Reviewer: 1

Reviewer Name: Margaret Cruickshank

This is a very important issue for all screening programmes - communication/engagement with the eligible population. Current programmes all depend on 19th century technology of the postal service and yet there are many groups within the population who are eligible for screening but have never used the postal service and use other means of communication.

The references are relevant and update and make the point clearly from the STRATEGIC trial that although internet booking was used (and should some improvement in screening uptake in younger women at first invitation), the participants were informed of this service by post.

We have undertaken qualitative research in this area and found similar issues and concerns regarding alternative means of communication.

The authors have used a different and interesting approach to access a wide range of women not just those accessing healthcare. They describe the limitations clearly but have still ended up with a demographic which tends to be more engaged in terms of age, education and deprivation. A means of

engaging with hard to reach groups is outwith the scope of this work and it still delivers an important message on communication.

Some of the concerns about confidentiality seem to ignore that a letter lying in a communal stairway is also easily accessible and safe delivery of mail cannot be guaranteed.

The authors discuss the proportion of GP practices in England which allow internet booking. Is this available in other countries and how generalizable are these results in terms of communication used in screening programmes and uptake of screening?

We have updated the title to highlight that this work was carried out in Great Britain. We have also outlined in the Discussion that these findings may not be generalisable to screening programmes in other countries.

Reviewer: 2

Reviewer Name: Alastair Gray

1) The survey was conducted as part of a national omnibus study. I didn't see much discussion of whether those surveyed are representative nationally of that age/gender group, in terms of employment, socio-economic status, ethnicity etc.

We have added some comments to the Discussion related to the representativeness of the sample.

2) A limitation of the study design (acknowledged) is that the non-response/refusal rate is not known. Another (not acknowledged) is that actual participation in the screening programme is self-reported only, and may not be completely accurate. It would be useful to cite any literature comparing self-reported and actual participation.

We have updated the Discussion, citing relevant literature indicating that self-reported measures of screening status should be interpreted with caution.

3) The authors do not report in tabular form some of the key responses to the questions they used. For example, Have you ever been diagnosed with cervical cancer, Have you had cervical cancer screening 1-7 responses, will you go when next invited 1-4 responses. Also the number fo barriers is mentioned in summary but not tabulated in detail. A couple of summary remarks are made about these in the Results but much more detail could be given and discussed.

We have now included a flow diagram chart outlining the number of participants who had been diagnosed with cervical cancer, had a hysterectomy or didn't provide sufficient data to determine screening status (Online Supplement 2). We have also provided more written text regarding barriers to participation in the Results section, to accompany Table 2.

4) How are Intenders actually defined? The text says "intended to be screened but were currently overdue". How was this constructed from the available questions? Are the yes probably and yes definitely grouped together? What are the proportions of these?

We have provided additional information in Online Supplement 1 and Online Supplement 2 to outline how 'overdue' and 'up-to-date' women were defined and to clarify that those who responded 'yes, probably' and 'yes, definitely' were grouped together. We have also given numbers of 'yes, definitely' and 'yes, probably' in Online Supplement 2.

5) The way other responses are dichotomised is described in a confusing way: for example, if a mode of communication is deemed acceptable: it seems to imply that acceptable includes quite or very acceptable and unacceptable includes quite/very unacceptable but also neither unacceptable or acceptable, don't know or not applicable. Same with Likelihood of using a method of booking. Is this just poorly described? And how many actually were in the neither one nor another, or don't know categories. And what does not applicable mean, if these are hypothetical questions about acceptability of different things?

We have updated the Methods section, to further clarify how responses were coded.

6) All results are presented in univariate tables. It would be worth conducting and reporting a multivariate analysis as some of these variables are likely to be strongly correlated.

We had conducted and reported multivariate analyses in Online Supplement 2 (now Online Supplement 3). These are also described and referenced in the results section.

7) The statement in the last paragraph of the discussion that because preferences did not seem to vary much by screening status, this negates ANY concern that changing the current screening programme might deter current participants, seems very strong and should at least be qualified.

We have updated this paragraph in the Discussion and commented that “The lack of differences by screening status suggests that changing the architecture should not deter ‘maintainers’ from participation.”

8) PPI statement: should this say both up-to-date and overdue?

We thank the reviewer for highlighting this typo. We have updated the PPI statement to correct this error.

Reviewer: 3

Reviewer Name: Jasmin Tiro

This study surveyed British women to understand practical barriers to booking a cervical cancer screening appointment and identify preferred booking modes. This was a novel exploration of the intention-behavior gap and investigators used the TRIALS framework to identify which problems most interfere with the desired action (booking an appointment). Other strengths include the representative sample, comprehensive measurement of practical barriers, and stratified analysis by past behavior (maintainers vs. intenders). The main weakness of the paper was whether the survey was informed by implementation science and engaged the perspectives of general practice and National Health Service stakeholders. It was unclear whether all invitation and booking options are feasible to implement and should have been offered to women. Specific questions and suggestions are noted below.

1. Abstract, conclusion: what do you mean by tailoring? Will you ask GPs to note each woman's preference and then match the appointment booking process to their preferred method? Or are you just trying to determine which options should be offered to women?

We have updated the Abstract to say “Women who are overdue for screening face practical barriers to booking appointments. Future interventions may assess the efficacy of changing the architecture of

the invitation and booking system. This may help women overcome logistical barriers to participation and increase coverage for cervical screening.”

2. Introduction: can you say more specifically what attributes of the alternative booking methods address problems highlighted by Sheeran/Webb’s TRIALS framework? Similarly, can you interpret your findings with those attributes in mind? For example, were you surprised that more women didn’t prefer the 24 hour automated service or call back options.

We have updated the Introduction to provide an example of how alternative booking methods may help address problems highlighted by the TRIALS model and provided further commentary in the Discussion to speculate reasons for preferences.

3. Methods: did you engage GP and NHS stakeholders in which alternative booking methods are feasible to implement (e.g., have resources, capacity)? If not, then this needs to be acknowledged in the Discussion and future studies should explore these implementation science issues or consider hybrid-effectiveness designs when testing proposed interventions (e.g., text messages during open GP hours).

We spoke with NHS stakeholders prior to conducting the study to ensure that the booking options outlined in the survey were considered feasible within the cervical screening programme in Great Britain. We have now included this in the Methods section and updated the Discussion, to say that further research is needed to determine how best to implement any changes to the screening programmes in Great Britain.

4. Discussion: Besides making it easier to quickly act in booking an appointment, what are other behavioral determinants that could be addressed to respond to “forgetting to book” (failing to start)? Should investigators try to enhance urgency or motivation?

This is an interesting research question; we have updated the Discussion to comment that further research exploring the content of the letter may be needed to increase the salience among screening invitees.

VERSION 2 – REVIEW

REVIEWER	Margaret Cruickshank Aberdeen Centre for Women's Health Research University of Aberdeen UK
REVIEW RETURNED	26-Apr-2019

GENERAL COMMENTS	My previous comments have been addressed by the authors
---

REVIEWER	Alastair Gray University of Oxford, UK
REVIEW RETURNED	11-Apr-2019

GENERAL COMMENTS	In general I am happy with the responses and changes to the manuscript. Concerning the questions on whether different modes of communication were acceptable or not, the classification has now been clarified but I still think that it is odd to classify "neither unacceptable or acceptable", "don't know" and "not applicable" as
---

	unacceptable. I would be reassured to see some more information on a) how many responses were in these categories and b) a sensitivity analysis reporting what difference it makes to the main findings if the invitation preferences are analysed solely on the basis of those who said something was quite or very acceptable, or quite or very unacceptable. I've marked the absence of this information as a study limitation at present.
--	---

VERSION 2 – AUTHOR RESPONSE

Reviewer's comments:

I still think that it is odd to classify "neither unacceptable or acceptable", "don't know" and "not applicable" as unacceptable. I would be reassured to see some more information on a) how many responses were in these categories and b) a sensitivity analysis reporting what difference it makes to the main findings if the invitation preferences are analysed solely on the basis of those who said something was quite or very acceptable, or quite or very unacceptable. I've marked the absence of this information as a study limitation at present.

We have now included additional descriptives tables in Online Supplement 3, reporting how many responses were in each category (in line with the reviewer's suggestion).

We re-ran the logistic regression analyses excluding those who responded "don't know" and "not applicable". There was very little difference in the findings, however we have decided to edit the results with these participants excluded because we acknowledge that classifying these two categories as 'unacceptable' or 'unlikely' may not be appropriate.

We believe that analyses comparing those who selected acceptable (quite or very) with a combined reference group including those who selected unacceptable (quite or very) and "neither unacceptable or acceptable" is appropriate, so we have left these participants in the reference group. However, we agree that the way we have described these groups may have been misleading and we have edited the manuscript throughout so that the groups are referred to as: acceptable v unacceptable/ambivalent (for invitation preferences) and likely to use v unlikely to use/ambivalent (for booking preferences).

VERSION 3 – REVIEW

REVIEWER	Alastair Gray University of Oxford
REVIEW RETURNED	29-May-2019
GENERAL COMMENTS	Thank you for making these changes. I believe these improve the paper and am happy to now see it published.